# Preoperative C-Reactive Protein-to-Albumin Ratio and Its Ability to Predict Outcomes of Pancreatic Cancer Resection: A Systematic Review

**DOI:** 10.3390/biomedicines11071983

**Published:** 2023-07-13

**Authors:** Shahin Hajibandeh, Shahab Hajibandeh, Saleh Romman, Alessandro Parente, Richard W. Laing, Thomas Satyadas, Daren Subar, Somaiah Aroori, Anand Bhatt, Damien Durkin, Tejinderjit S. Athwal, Keith J. Roberts

**Affiliations:** 1Department of Hepatobiliary and Pancreatic Surgery, Royal Stoke University Hospital, University Hospitals of North Midlands NHS Trust, Stoke-on-Trent ST4 6QG, UK; saleh.romman23@gmail.com (S.R.); richwlaing@gmail.com (R.W.L.); anand.bhatt@uhnm.nhs.uk (A.B.); damien.durkin@uhnm.nhs.uk (D.D.); tejinderjit.athwal@uhnm.nhs.uk (T.S.A.); 2Department of Hepatobiliary and Pancreatic Surgery, University Hospital of Wales, Cardiff CF14 4XW, UK; shahab_hajibandeh@yahoo.com; 3Division of Hepatobiliary and Liver Transplantation, Department of Surgery, Asan Medical Center, University of Ulsan College of Medicine, Seoul 05505, Republic of Korea; aleparen@gmail.com; 4Department of Hepatobiliary and Pancreatic Surgery, Manchester Royal Infirmary Hospital, Manchester M13 9WL, UK; tomsaty@yahoo.co.uk; 5Department of Hepato-Pancreato-Biliary Surgery, Royal Blackburn Hospital, Blackburn BB2 3HH, UK; daren.subar@elht.nhs.uk; 6Department of HPB Surgery, University Hospitals Plymouth NHS Trust, Plymouth PL6 8DH, UK; somaiah.aroori@gmail.com; 7Department of Hepato-Pancreato-Biliary and Liver Transplant Surgery, Queen Elizabeth University Hospitals Birmingham NHS Foundation Trust, Birmingham B15 2TH, UK; keith.roberts@uhb.nhs.uk

**Keywords:** pancreatic cancer, C-reactive protein, Albumin, prognostic score

## Abstract

Objectives. To evaluate the ability of the c-reactive protein-to-albumin ratio (CAR) in predicting outcomes in patients undergoing pancreatic cancer resection. Methods. A systematic search of electronic information sources and bibliographic reference lists was conducted. Survival outcomes and perioperative morbidity were the evaluated outcome parameters. Results. Eight studies reporting a total of 1056 patients undergoing pancreatic cancer resection were identified. The median cut-off value for CAR was 0.05 (range 0.0003–0.54). Using multivariate analysis, all studies demonstrated that a higher CAR value was an independent and significant predictor of poor overall survival in patients undergoing pancreatic cancer resection. The estimated hazard ratio (HR) ranged from 1.4 to 3.6. Although there was a positive correlation between the reported cut-off values for CAR and HRs for overall survival, it was weak and non-significant (r = 0.36, *n* = 6, *p* = 0.480). There was significant between-study heterogeneity. Conclusions. Preoperative CAR value seems to be an important prognostic score in predicting survival outcomes in patients undergoing pancreatic cancer resection. However, the current evidence does not allow the determination of an optimal cut-off value for CAR, considering the heterogeneous reporting of cut-off values by the available studies and the lack of knowledge of their sensitivity and specificity. Future research is required.

## 1. Introduction

Pancreatic cancer is the fourth-leading cause of cancer-related mortality worldwide [1]. Its associated 5-year survival with consideration of all stages of the disease is estimated to be 9% [1]. Even though a surgical resection of pancreatic cancer is possible, 5-year survival after surgical resection remains between 18% and 31.6% despite the recent advances in preoperative diagnosis, surgical techniques, and peri-operative care [2,3,4].

Over the last two decades, more than 20 prognostic models for resectable pancreatic cancer have been developed to predict survival outcomes. However, more than 90% of the prediction models have never been validated externally, and those that have been validated either internally or externally have had sub-optimal performance [5]. Furthermore, most of the models did not consider any inflammation-based score.

The systemic inflammatory response has been demonstrated to play a critical role in carcinogenesis and tumor progression [6,7], subsequent cancer-related malnutrition [8], and survival [9,10,11]. Several inflammation-based scores, including modified Glasgow prognostic score (mGPS) [9,10,11] and neutrophil-to-lymphocyte ratio (NLR) [12,13], have been validated to be associated with tumor progression and prognosis in various cancers, including pancreatic cancer. More recently, the C-reactive protein (CRP) to albumin ratio (CAR) has been demonstrated to be associated with survival outcomes in cancer patients [14,15]. Although two recent meta-analyses [16,17] demonstrated that CAR was associated with the survival of pancreatic cancer patients, the validity of their findings is of doubtful merit as the authors in both studies conducted pooled analysis from studies with heterogeneous cut-off values for CAR. In the presence of various cut-off values, the conduct of a pooled analysis of interested outcomes is not only not indicated but also contraindicated, as the patients in high and low CAR groups in one study can meet the inclusion criteria for low and high groups in other studies, respectively, and vice versa. Moreover, both meta-analyses included resectable and non-resectable cancers together, did not evaluate important baseline characteristics of the included patients, and missed several eligible studies. Therefore, a robust systemic evaluation of the real evidence surrounding the ability of CAR to predict survival outcomes in patients with resectable pancreatic cancer is lacking. 

A novel and robust systematic review should focus on only the preoperative value of CAR in resectable pancreatic cancer, assess the reliability of the provided cut-off values reported by the best available evidence, and provide strategies to improve the quality of the evidence in order to prepare CAR for potential inclusion in future prediction models for pancreatic cancer. We aimed to conduct a comprehensive systematic review of all available comparative studies to evaluate the ability of CAR to predict outcomes in patients undergoing pancreatic cancer resection.

## 2. Methods

### 2.1. Design and Study Selection

The inclusion and exclusion criteria, methodology, and investigated outcome parameters of this review were highlighted in a review protocol. We registered our protocol at the International Prospective Register of Systematic Reviews (registration number: CRD42023440251). Our methodology respected the standards of the preferred reporting items for systematic reviews and meta-analyses (PRISMA) statements [18].

### 2.2. Types of Studies

Considering that CAR is not an intervention, performing a randomized controlled trial (RCT) on the topic of this study is not possible. Therefore, all comparative observational studies investigating the outcomes of pancreatic cancer resection in patients with preoperative high or low CAR were included. Only studies that determined a cut-off value for CAR and classified the patients into a high or low CAR group were considered. Considering that this study aimed to synthesize comparative evidence by directly comparing the outcomes of high CAR and low CAR, all non-comparative observational studies, review articles, letters or expert opinions, correspondence, and editorials were excluded. Moreover, studies that classified their patients into a high or low CAR group based on postoperative values were excluded.

### 2.3. Types of Participants

Any adult patient (aged 18 or over) of any gender who underwent open, laparoscopic, or robotic pylorus-preserving pancreaticoduodenectomy (PPPD), pancreaticoduodenectomy (PD), distal pancreatectomy (DP), central pancreatectomy (CP), or total pancreatectomy (TP) for malignant disease of the pancreas was considered for inclusion. Patients who had the aforementioned procedures for benign or malignant pathologies, including cholangiocarcinoma, duodenal cancer, or ampullary cancer, were excluded. 

### 2.4. Outcomes

We planned to evaluate the outcome parameters, including survival outcomes including overall survival (OS) (time-to-event), 1-year OS, 2-year OS, 4-year OS, 5-year OS, and disease-free survival (DFS), and peri-operative (within 30 days of operation) morbidity (classified as Clavien–Dindo less than 3 or Clavien–Dindo equal or more than 3). 

### 2.5. Literature Search Strategy

We created a robust search strategy and conducted the literature search in MEDLINE, EMBASE, CINAHL, and CENTRAL. There were two independent authors involved in this stage. The same authors searched http://apps.who.int/trialsearch/ (accessed on 10 March 2023), http://clinicaltrials.gov/ (accessed on 10 March 2023), and http://www.isrctn.com/ (accessed on 10 March 2023) to look for incomplete and unpublished studies. Furthermore, the reference lists of the included articles and the previous review articles were assessed for potentially eligible studies. The literature search began on 10 March 2023 and lasted for three days. Appendix B presents the search strategy that was used for the literature search.

### 2.6. Selection of Studies

Following the literature search, two authors evaluated the details of the identified articles, including title, abstract, or full text, according to the inclusion and exclusion criteria of this study. Those that met our inclusion criteria were included. The same authors resolved disagreements in this process via discussion. However, in cases of persistent discrepancies, a third author was consulted.

### 2.7. Data Extraction and Management

With respect to Cochrane’s recommendations, two independent reviewers evaluated the included studies and extracted:Study-related data (first author, publication year, country of origin of the corresponding author, journal in which the study was published, study design, procedure performed, and sample size in each group)Baseline demographic and clinical information of the study populations (age, gender, American Society of Anesthesiologists (ASA) grade, cut-off CAR value and its associated sensitivity, specificity, positive predictive value (PPV), and negative predictive value (NPV), use of neoadjuvant chemotherapy, use of adjuvant chemotherapy, pathological tumor-node-metastasis (pTNM) staging, tumor site and size, vascular involvement, lymph node metastasis, preoperative cancer antigen 19-9 (CA 19-9), carcinoembryonic antigen (CEA), neutrophil-to-lymphocyte ratio (NLR), and modified Glasgow prognostic score (mGPS) 1 or 2)Outcome data

Discrepancies during this process were resolved following consultation with an additional author. 

### 2.8. Assessment of Risk of Bias

As all of the included studies in this review were observational studies, we assessed their methodological quality and risk of bias against criteria highlighted in the Newcastle–Ottawa scale (NOS) [19]. Two authors were involved. We resolved discrepancies in the risk of bias assessment through discussion between the assessing authors. Nevertheless, if no agreement could be reached, a third reviewer was involved as an adjudicator.

### 2.9. Summary Measures and Synthesis

In the pre-defined protocol, we planned to conduct a meta-analysis of dichotomous and continuous outcome parameters. However, as the included studies reported heterogeneous cut-off values for CAR, the conduct of pooled analyses for our defined outcome measures was deemed inappropriate. A Pearson correlation coefficient was calculated to assess the relationship between the CAR cut-off values and the reported hazard ratio (HR) for OS. A two-sided confidence interval with a 95% confidence level was used to indicate statistical significance.

## 3. Results

Our literature search resulted in 1577 articles. After further evaluation of the identified articles, 22 articles were shortlisted for potential inclusion. A further 14 studies were excluded, as 5 were single-arm studies, 2 evaluated postoperative values, and the remaining 7 included patients who did not have pancreatic cancer resection. Therefore, 8 retrospective comparative studies [20,21,22,23,24,25,26,27] were deemed appropriate for inclusion (Figure 1). The included studies reported a total of 1056 patients undergoing pancreatic cancer resection, of whom 378 belonged to the high CAR group and the remaining 678 belonged to the low CAR group.

Table 1 presents the date of publication and country of origin, journal, and study design of the included studies, performed procedures, sample sizes, and cut-off values in each study. Table 2 and Table 3 present the baseline characteristics of the study populations. Moreover, Table 4 presents a summary of the included studies.

Cut-off CAR values. All included studies provided cut-off values for CAR to define their study groups using receiver operating characteristic (ROC) analysis. The median cut-off value for CAR was 0.05 (range 0.0003–0.54). Of the included studies, only Wijk et al. [21] reported the sensitivity, specificity, PPV, and NPV of their reported cut-off value, which were 54%, 69%, 78%, and 42%, respectively. 

Performed procedure. Four studies [20,21,22,27] specifically reported the names of performed procedures in their study groups, which included 52% PDs, 27% PPPDs, 14% DPs, 6% TPs, and 1% CPs in the high CAR group, and 41% PDs, 27% PPPDs, 26% DPs, 1% TPs, and 5% CPs in the low CAR group. 

Neoadjuvant chemotherapy. Five studies [20,21,22,24,25] provided information about the use of neoadjuvant chemotherapy in their high and low CAR groups. Although the use of neoadjuvant chemotherapy was comparable between the high and low CAR groups in each study, there was significant heterogeneity among the included studies. Murakawa et al. [22] and Ikeguchi et al. [25] reported no use of neoadjuvant chemotherapy in their included patients. Oshima et al. [20] reported the use of neoadjuvant chemotherapy in all patients included. Wijk et al. [21] reported the use of neoadjuvant chemotherapy in 97% and 96% of patients in the high and low CAR groups, respectively. Ikuta et al. [24] reported the use of neoadjuvant chemotherapy in 33% of patients in the high CAR group and 25% of patients in the low CAR group. 

Adjuvant chemotherapy. Four studies [20,21,22,24] reported the use of adjuvant chemotherapy in their study groups. The rate of adjuvant chemotherapy was comparable between the high and low CAR groups in each study, while it was heterogeneously reported among the included studies. Murakawa et al. [22] reported the use of adjuvant chemotherapy in all included patients. The rates of adjuvant chemotherapy in the high and low CAR groups were 38% vs. 31% in the study of Wijk et al. [21], 54% vs. 42% in the study of Oshima et al. [20], and 77% and 84% in the study of Ikuta et al. [24].

pTNM staging. Four studies [22,24,25,27] reported the pTNM staging of the included patients in the high and low CAR groups. The pTNM stages included were stage I and II in the study of Murakawa et al. [22], stages I–III in the study of Ikeguchi et al. [25], and stage I–IV in the studies of Ikuta et al. [24] and Haruki et al. [27]. In all the included studies, the patients in the high CAR group had a more advanced pTNM stage.

Lymph node metastasis. Four studies [20,21,22,27] provided information about the lymph node metastasis of their included patients. Wijk et al. [21] reported 100% lymph node metastasis in both high and low CAR groups. The authors classified their patients into <5 and ≥5 lymph node metastases and found no significant difference in the proportion of patients in the high and low CAR groups (*p* = 0.183). Oshima et al. [20] reported the presence of lymph node metastasis in 53.8% and 50.0% of patients in the high and low CAR groups, respectively, with no significant difference between the two groups (*p* = 0.8139). Similarly, Ikuta et al. [24] found no significant difference in lymph node metastasis between the two groups (86.7% vs. 72.6%, *p* = 0.15). Murakawa et al. [22] reported a significantly higher rate of lymph node metastasis in the high CAR group when compared with the low CAR group (81.2% vs. 58%, *p* = 0.008). 

Preoperative CA19-9 level. Four studies [20,21,23,24] reported the level of preoperative CA19-9 in the high and low CAR groups. However, the way that CA19-9 levels are reported varies significantly among the included studies. Wijk et al. [21] and Oshima et al. [20] reported the median CA19-9 levels (with and without range, respectively) in the high and low CAR groups and did not find any significant difference between the two groups. Vujic et al. [23] reported the preoperative CA19-9 level in their study groups as a mean (without standard deviation (SD) of the mean) and did not find any significant difference in preoperative CA19-9 between the high and low CAR groups. Ikuta et al. [24] reported the CA19-9 as categorical data after determining a cut-off value of 105 U/mL and comparing the number of patients in each category between the high and low CAR groups. The authors found no significant difference between the two groups. 

Preoperative CEA level. Three studies [20,21,23] reported the level of preoperative CEA in the high and low CAR groups. However, the way that CA19-9 levels are reported varies significantly among the included studies. Wijk et al. [21] and Oshima et al. [20] reported the median CEA levels in their study groups, and in both studies, there was no significant difference in the preoperative CEA level between the high and low CAR groups. Vujic et al. [23] reported the mean CEA (without the SD of the mean) in the high and low CAR groups and found no significant difference between the two groups.

NLR. The included studies poorly reported the preoperative NLR in their study groups. Only two studies [20,24] provided such values for their high and low CAR groups, and in both studies, there was no significant difference in NLR between the two groups. 

mGPS 1 or 2. Only three studies [20,24,27] provided information about the preoperative mGPS 1 or 2 in the high and low CAR groups. There was a significantly higher number of patients with mGPS 1 or 2 in the high CAR group in the studies of Ikuta et al. [24] (16.7% vs. 0%, *p* < 0.001) and Haruki et al. [27] (32.7% vs. 1.8%, *p* < 0.001) when compared to the low CAR group. Furthermore, although there was a higher number of patients with pGPS 1 or 2 in the high CAR group in the study of Oshima et al. [20], the difference did not reach statistical significance (15.4% vs. 8.3%, *p* = 0.47).

The included studies poorly reported tumor site, size, and vascular involvement in their study groups. Nevertheless, those that reported the aforementioned characteristics demonstrated the comparability of the study groups in each study.

### 3.1. Methodological Appraisal

The methodological appraisal of all eight included observational studies is presented in Appendix A. The risk of bias was judged moderate in seven studies and high in one study.

### 3.2. Outcome Data 

Outcomes are summarized in Table 5.

### 3.3. Postoperative Complications

Four studies [20,21,22,27] provided information about postoperative complications in their study groups using the Clavien–Dindo classification. In all five studies, there was no significant difference in Clavien–Dindo grades 0–2 and Clavien–Dindo grades 3–5 between the high and low CAR groups. The rate of Clavien–Dindo grades 3–5 in the high CAR group ranged from 15% to 28%, which was comparable to that in the low CAR group (16% to 28%). Procedure-specific complications, such as POPF, were poorly reported in the included studies.

### 3.4. Overall Survival

All included studies [20,21,22,23,24,25,26,27] reported the outcomes of survival analysis with respect to their determined cut-off CAR values using univariate and multivariate analyses. The estimate of effect size was reported as the hazard ratio (HR) in six studies and the odds ratio (OR) in two studies. All included studies demonstrated that a higher CAR was an independent and significant predictor of poor overall survival. The estimated HR provided by the included studies ranged from 1.4 to 3.6. A Pearson correlation coefficient was calculated to assess the relationship between the CAR cut-off values and the reported HR for OS. Although there was a positive correlation between the two variables, it was weak and non-significant (r = 0.36, *n* = 6, *p* = 0.480) (Figure 2). 

### 3.5. 1–5 Year Survival Rates

Wijk et al. [21] reported 1–4 year survival rates of patients in the high and low CAR groups. Although the 1-year survival rate was comparable in both groups (84% vs. 87%), the high CAR group had lower 2 year-survival (67% vs. 73%), 3-year survival (50% vs. 60%), and 4-year survival (34% vs. 46%) rates than the low CAR group. However, the authors did not provide any *p*-values to determine statistical significance. Vujic et al. [23] reported 1–5 year survival rates of patients in the high and low CAR groups. The authors demonstrated significantly lower 1-year survival (56% vs. 67%, *p* = 0.013), 2 year-survival (29% vs. 35%, *p* = 0.013), 3-year survival (12% vs. 21%, *p* = 0.013), 4-year survival (8% vs. 15%, *p* = 0.013), and 5-year survival (3% vs. 7%, *p* = 0.013) rates than the low CAR group. Murakawa et al. [22] only reported 5-year survival in their study groups and demonstrated that high CAR was associated with significantly lower 5-year survival when compared with low CAR (22.5% vs. 36.4%, *p* = 0.0089). 

### 3.6. Disease-Free Survival 

Only three studies [22,24,27] provided information about DFS in their included patients. Ikuta et al. [24] demonstrated that patients with high CAR had a significantly shorter median disease-free time than patients in the low CAR groups (9.3 months vs. 22.1 months, *p* < 0.001). Murakawa et al. [22] demonstrated that patients in the high CAR group had significantly lower 5-year DFS rates compared to those in the low CAR group (12.5% versus 22.1%, *p* = 0.0097). Haruki et al. [27] found significantly lower DFS in the high CAR group than in the low CAR group (*p* = 0.049).

## 4. Discussion

Considering that chronic inflammation has a direct causal relationship with carcinogenesis and the fact that malignancy itself stimulates an inflammatory response leading to deleterious effects on the malignant process [28], the interest in inflammation-based prognostic scores such as the NLR, GPS, mGPS, and, more recently, CAR has increased in many types of cancer [28,29,30,31,32].

Given recent debates regarding a comprehensive systematic review of eight available comparative observational studies [20,21,22,23,24,25,26,27], enrolling a total of 1056 patients undergoing pancreatic cancer resection was conducted. Out of 1056 patients, 378 belonged to the high CAR group, and the remaining 678 belonged to the low CAR group. The subsequent systematic evaluation of the baseline characteristics of the included patients and the reported outcome data demonstrated that although all the included studies are in agreement that a higher CAR value was an independent and significant predictor of poor OS, an optimal cut-off value cannot be determined based on the current evidence. Moreover, despite the existence of within-study homogeneity, there was significant between-study heterogeneity regarding the performed procedures, the use of neoadjuvant or adjuvant chemotherapy, pTNM staging, lymph node metastasis, preoperative CA19-9 and CEA levels, NLR, and mGPS. Furthermore, some important prognostic characteristics, including tumor site and size, pathological T-factor, and vascular involvement, were poorly reported by most of the included studies.

CRP is an acute-phase reactant that is mainly produced by hepatocytes under the influence of proinflammatory cytokines, particularly interleukin-6 (IL-6), which are commonly over-produced in cancer cells and immune cells that infiltrate tumor tissue [25,33]. Increased serum CRP levels have been associated with poor outcomes in most gastrointestinal malignancies, including pancreatic cancer [34,35,36]. Serum albumin is a negative acute-phase reactant that is closely associated with inflammation [25]. Although the underlying cause of low albumin levels in cancer patients remains unknown, it has been demonstrated that high levels of IL-6 produced by cancer cells inhibit the production of albumin [37]. Hypoalbuminemia is commonly associated with poor performance status, weight loss, and nutritional deficiency, which negatively impact the prognosis of cancer patients [38,39]. Both CRP and albumin are the main components of mGPS, which combine the increase in serum CRP level and the decrease in albumin concentration to provide scores between 0 and 2. Nevertheless, the usefulness of mGPS as a predictor of survival in pancreatic cancer has been controversial [21,26,27]. The main advantage of using CAR is that it can be considered a continuous variable in a multivariate analysis to demonstrate a correlation between CAR and survival. This enabled all of our included studies to find such a correlation. In fact, despite the existence of the aforementioned heterogeneous baseline characteristics, all the included studies conducted univariate and multivariate analyses, which demonstrated that a higher preoperative CAR is an independent predictor of poor OS in patients undergoing pancreatic cancer resection. 

The main limitation associated with the best available evidence in this context is the failure to provide the most accurate cut-off value for CAR. The selection of an appropriate cut-off value is of cardinal importance and is closely related to the sensitivity and specificity of that cut-off value on the ROC curve. Although a cut-off value is commonly determined at a point where sensitivity equals specificity, sometimes, depending on the nature of the condition of interest, the former can be compromised to enhance the latter, and vice versa [40]. Only one of the included studies reported the sensitivity and specificity associated with the provided cut-off value (cut-off value: 0.2), which were 54% and 69%, respectively [21]. Therefore, the most sensitive and specific preoperative cut-off value for CAR in patients undergoing pancreatic cancer resection remains unknown. Without defining the most sensitive and specific cut-off value for CAR, escalation of the level of evidence in this context would be challenging. Considering that CAR is not an intervention, designing an RCT is not possible. However, a prospective cohort study will be able to provide stronger evidence for defining high CAR groups (the exposed group) and low CAR groups (the non-exposed group) if an appropriate cut-off value is determined.

We did not conduct any meta-analysis of the outcomes and baseline characteristics. In the absence of a robust cut-off value for CAR, it is important to avoid any pooled analysis during evidence synthesis, as such analyses can potentially lead to misleading conclusions. This is because the patients in high and low CAR groups in one study can meet the inclusion criteria for low or high groups in other studies. Recently, Fu et al. [16] and Zang et al. [17] conducted a meta-analysis of 11 and 9 studies, respectively, enrolling only Asian patients with pancreatic cancer and concluded that CAR is associated with the survival of pancreatic cancer patients of Asian ethnicity and that a higher CAR may be a potential prognostic indicator in pancreatic cancer. However, there are several important concerns about their meta-analysis, in addition to the aforementioned concern about the heterogeneous cut-off values that both meta-analyses pooled together. The authors included studies that did not consider surgical resection and treated the patients with chemotherapy alone. Moreover, Fu et al. [16] included studies that considered postoperative CAR values rather than preoperative CAR. It should be taken into account that although postoperative values may still have some prognostic significance, they reflect not only cancer progression but also surgical invasion. Pancreatic cancer resection is one of the most invasive surgical procedures that can cause systemic inflammation, postoperative complications, and poor nutritional status [22]. Finally, both meta-analyses missed several relevant studies meeting their inclusion criteria. 

Several factors have been identified as predictors of survival outcomes after pancreatic cancer resection, and it may be impossible to make a prognostic model based on all identified prognostic factors. In 2004, a prognostic nomogram for patients undergoing resection for adenocarcinoma of the pancreas was developed in order to serve as a basis for investigating other potentially predictive variables that are proposed to be of prognostic importance [41]. Since then, several prediction models for resectable pancreatic cancer have been developed with poor model performance. In a comprehensive review, Strijker et al. [5] systematically evaluated 21 prediction models, which investigated common prognostic factors such as biomarkers (such as CA19.9 and albumin) and pathological factors (differentiation grade, nodal status, tumor size, and margin status). The authors demonstrated that of the 19 models developed or updated, only two underwent formal external validation. Interestingly, none of the applicable models that underwent internal or external validation had an area under the curve (AUC) value above 0.7. Moreover, the authors detected a high risk of bias in most studies [5]. This may be the time to introduce valid inflammation-based prognostic factors into potential future prediction models for resectable pancreatic cancer. A balanced model should consist of sensitive and specific clinico-pathological, inflammation, and nutrition-based prognostic scores. Several other cytokines or inflammatory markers can potentially be considered representatives of cancer-induced inflammation in patients with pancreatic cancer. Nevertheless, the available literature on other inflammatory markers is even more limited than CAR. One of the advantages of CAR is that its components are routinely reported in the blood results of almost all patients presenting to secondary/tertiary care. Among inflammation-based prognostic scores, CAR and NLR have the greatest potential to represent the degree of inflammation in any future predictive model. Nevertheless, the introduction of such scores, particularly regarding CAR, needs patience. The level and quality of evidence need to be improved regarding the predictive ability of CAR in patients undergoing pancreatic cancer resection. This study is the first review to comprehensively evaluate the best available evidence surrounding the ability of CAR to predict the outcomes of pancreatic cancer resection and recognize the shortcomings of such evidence. Considering the findings of this review, we have the following novel directions for future research to enhance the quality of evidence and provide appropriate grounds for higher-level evidence: Consideration of preoperative values for CAR rather than postoperative values.Reporting of the sensitivity and specificity of any determined cut-off value.Providing the AUC value of the determined cut-off values.Reporting of CAR as continuous data (mean ± SD or median (IQR)) in both high and low CAR groups so that the aforementioned data can be used for a meta-analysis or pooled analysis.Consideration of DFS, mean survival time (MST), and disease-specific survival alongside overall survival as outcome parameters.Inclusion of a large sample size to minimize the risk of type 2 error.

The limitations of this review should be taken into account when interpreting its findings. All of the included studies were retrospective observational studies, which are subject to selection bias. All of the included studies had a moderate or high risk of bias. Most importantly, we could not conduct a meta-analysis of outcomes due to the heterogeneously reported cut-off values for CAR.

## 5. Conclusions

The findings of this systematic review indicate that the preoperative CAR value seems to be an important prognostic score in predicting survival outcomes in patients undergoing pancreatic cancer resection. However, the current evidence does not allow the determination of an appropriate cut-off value for CAR, considering the heterogeneous reporting of cut-off values by the available studies and the lack of knowledge of their sensitivity and specificity. We encourage future studies to consider the directions for future research that this review recommended to provide stronger evidence in favor of introducing preoperative CAR into prediction models for pancreatic cancer resection, which have been lacking an inflammation-based score until today.

## Figures and Tables

**Figure 1 biomedicines-11-01983-f001:**
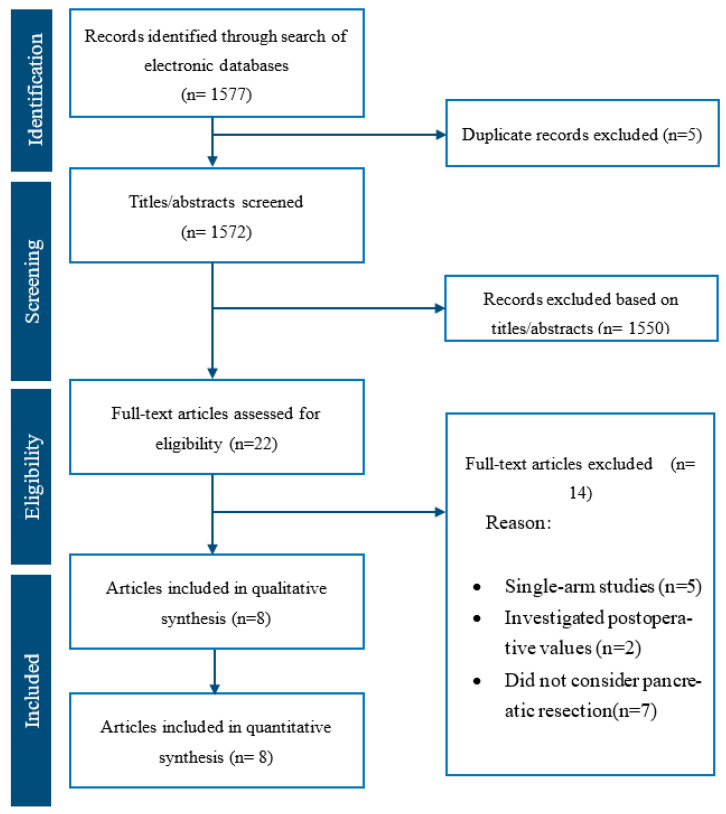
PRISMA flow chart.

**Figure 2 biomedicines-11-01983-f002:**
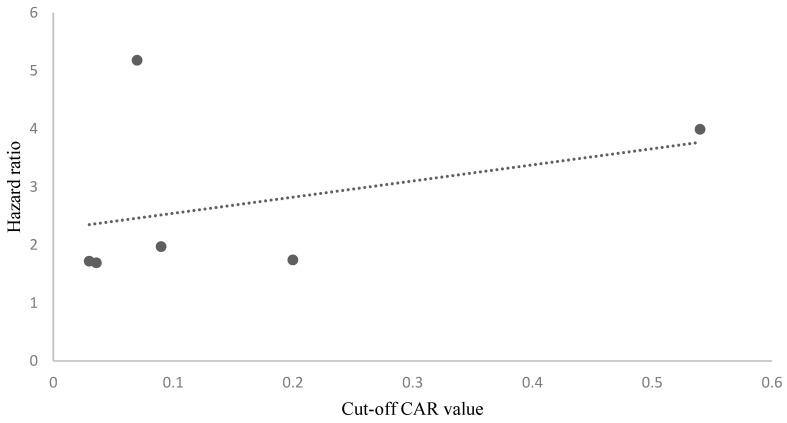
The scatterplot demonstrating a positive but non-significant correlation between the CAR cut-off values and the reported HRs for OS was weak and non-significant (r = 0.36, *n* = 6, *p* = 0.480).

**Table 1 biomedicines-11-01983-t001:** Included studies related data.

Author	Year	Country	Journal	Study Design	Procedure (PPPD:PD:DP:CP:TP)	High CAR	Low CAR	Cut off CAR Value
Oshima [20]	2020	Japan	BMC Gastroenterol	Retrospective observational study	0:34:11:0:4	13	36	0.07
Wijk [21]	2020	Netherlands	Eur J Med Res	Retrospective observational study	47:15:5:2: 4 vs. 59:9:17:0:5	73	90	0.2
Murakawa [22]	2020	Japan	In Vivo	Retrospective observational study	0:35:8:0:5 vs. 0:43:22:0:4	48	69	0.036
Vujic [23]	2019	Austria	In Vivo	Retrospective observational study	NR	59	143	0.0003
Ikuta [24]	2019	Japan	Asia Pac J Clin Oncol	Retrospective observational study	NR	30	106	0.09
Ikeguchi [25]	2017	Japan	Journal of Pancreatic Cancer	Retrospective observational study	NR	23	20	0.04
Wu [26]	2016	China	Tumor Biol	Retrospective observational study	NR	74	159	0.54
Haruki [27]	2016	Japan	World J Surg	Retrospective observational study	0:44:12:0:2 vs. 0:35:17:2:1	58	55	0.03

Abbreviations: PPPD, pylorus-preserving pancreaticoduodenectomy; PD, pancreaticoduodenectomy; DP, distal pancreatectomy; CP, central pancreatectomy; TP, total pancreatectomy; CAR, C-reactive protein to albumin ratio; NR, not reported.

**Table 2 biomedicines-11-01983-t002:** Baseline demographic and clinical characteristics of included studies.

Author	Age (Mean/Median)	Male: Female	ASA (1:2:3:4)	pTNM Stage	Tumour Site	Tumour Size	Lymph Node Metastasis	Vascular Involvement	Neoadjuvant Chemotherapy	Adjuvant Chemotherapy
Oshima [20]	74 vs. 68	7:6 vs. 20:16	NR	NR	Head/neck 73.5% Body/tail 26.5%	30 vs. 20, *p* = 0.076	53.8% vs. 50.0%, *p* = 0.8139	38.5% vs. 27.8. *p* = 0.4783	100% vs. 100%	54% vs. 42%
Wijk [21]	67 ± 9.7 vs. 65 ± 9.7	38:35 vs. 49:41	5:48:19:1 vs. 7:73:19:0	NR	NR	35 (25–40) vs. 30 (25–40), *p* = 0.477	<5 41% vs. 59% ≥5 53% vs. 47% *p* = 0.183	NR	97% vs. 96%	38% vs. 31%
Murakawa [22]	71 (46–81) vs. 68 (44–84)	22:26 vs. 41:28	NR	I 0% vs. 8.7% IIA 25% vs. 24 34.8% IIB 75% vs. 56.5%	NR	35 (15–83) vs. 32 (9–105), *p* = 0.568	81.2% vs. 58%, *p* = 0.008	NR	0% vs. 0%	100% vs. 100%
Vujic [23]	65.1 vs. 65.1	NR	27:32 vs. 76:67	NR	NR	NR	NR	NR	NR	NR
Ikuta [24]	NR	21:9 vs. 55.51	NR	I–II 56.7% vs. 77.4% III–IV 43.3% vs. 22.6%	Head 63.3 vs. 60.4 Body/tail 36.7 vs. 39.6 *p* = 0.94	NR	86.7% vs. 72.6%, *p* = 0.15	NR	33% vs. 25%	77% vs. 84%
Ikeguchi [25]	NR	NR	NR	IA 7% IB 4.7% IIA 16.3% IIB 58.1% III 14%	NR	NR	NR	NR	0% vs. 0%	NR
Wu [26]	NR	55:19 vs. 101:58	NR	NR	Head 50.0% vs. 42.1% Body 28.4% vs. 27.1% Tail 17.6% vs. 15.1% Diffusion 4.0% vs. 15.7% *p* = 0.084	NR	NR	NR	NR	NR
Haruki [27]	66.6± 10.0 vs. 66.9 ±11.1	36:22 vs. 34:21	NR	I 0% vs. 5.4% II 6.8% vs. 27.2% III 51.7% vs. 45.4% IV 34.5% vs. 16.4%	NR	NR	NR	NR	NR	NR

Abbreviations: ASA, American Society of Anesthesiologists; pTNM, pathological tumor-node-metastasis; NR, not reported.

**Table 3 biomedicines-11-01983-t003:** Biochemical and inflammatory factors reported by the included studies.

Author	CEA *	CA19-9 *	NLR *	mGPS 1 or 2 *
Oshima [20]	6.5 vs. 3.5, *p* = 0.1333	455 vs. 107, *p* = 0.0728	2.51 vs. 2.00 *p* = 0.0852	15.4% vs. 8.3%, *p* = 0.4761
Wijk [21]	3.2 (2.2–7.5) vs. 4.5 (2.2–6.6), *p* = 0.577	133 (62–1092) vs. 342 (54–867), *p* = 0.800	NR	NR
Murakawa [22]	NR	NR	NR	NR
Vujic [23]	7.9 vs. 4.6, *p* = 0.254	2248.9 vs. 787.8, *p* = 0.339	NR	NR
Ikuta [24]	NR	≤105 53% vs. 42.5% >105 46.7% vs. 57.5% *p* = 0.31	≤5.1 93.3% vs. 97.2% >5.1 6.7% vs. 2.8% *p* = 0.31	16.7% vs. 0%, *p* < 0.001
Ikeguchi [25]	NR	NR	NR	NR
Wu [26]	NR	NR	NR	NR
Haruki [27]	NR	NR	NR	32.7% vs. 1.8%, *p* < 0.001

Abbreviations: CEA, carcinoembryonic antigen; CA19-9, preoperative cancer antigen 19-9; NLR, neutrophil-to-lymphocyte ratio; mGPS, modified. Glasgow prognostic score; NR, not reported. * High CAR versus Low CAR.

**Table 4 biomedicines-11-01983-t004:** Summary of the included studies.

Author	Summary of the Included Studies
Oshima [20]	A total of 49 patients with resectable or borderline resectable PDAC after NACRT were included. A cut-off value of 0.077 for CAR was determined. The group with CAR > 0.077 lost more body weight during NACRT (*p* = 0.03). Higher CAR was associated with significantly shorter overall survival.
Wijk [21]	A total of 163 patients with resected PDAC were included. A cut-off value of 0.2 for CAR was determined. CAR ≥ 0.2 was associated with decreased overall survival (16 vs. 26 months, *p* = 0.003). Higher CAR was an independent indicator of decreased overall survival.
Murakawa [22]	A total of 117 patients with resectable PDAC were included who had radiacal surgery with S1 adjuvant chemotherapy. A cut-off value of 0.036 for CAR was determined. The 5-year overall survival (OS) rates in the high- and low-ratio groups were 22.5% and 36.4%, respectively (*p* = 0.0089). The 5-year disease-free survival rates in the high- and low-ratio groups were 12.5% and 22.1%, respectively (*p* = 0.0097). Higher CAR was an independent prognostic factor of overall survival.
Vujic [23]	A total of 202 patients with resected PDAC were included. A cut-off value of 0.0003 for CAR was determined. CAR was an independent prognostic factor of overall survival in univariate and multivariate Cox regression analysis. Elevated CAR was associated with a higher median value of the Charlson Index, a higher Union for International Cancer Control) classification and increased carcinoembryonic antigen levels.
Ikuta [24]	A total of 136 patients with resected PDAC were included. A cut-off value of 0.09 for CAR was determined. High CAR was an independent predictor of poor overall survival (*p* = 0.03).
Ikeguchi [25]	A total of 43 patients with resected PDAC were included. A cut-off value of 0.04 for CAR was determined. High CAR was strong preoperative marker of poor prognosis independently of tumor stage.
Wu [26]	A total of 233 patients with PDAC were included with an unspecified proportion of resectable disease. A cut-off value of pre-treatment (surgical or chemotherapy) 0.54 for CAR was determined. CAR was identified as the only inflammation-based parameter with an independent prognostic ability in the multivariate analyses (*p* < 0.001).
Haruki [27]	A total of 113 patients with resected PDAC were included. A cut-off value of 0.03 for CAR was determined. Higher CAR (*p* = 0.023) was an independent and significant predictor of poor patient outcome.

Abbreviations: PDAC, pancreatic ductal adenocarcinoma; NACRT, Neoadjuvant chemoradiotherapy; CAR, C-reactive protein to albumin ratio.

**Table 5 biomedicines-11-01983-t005:** Survival data and perioperative morbidities reported by the included studies.

Author	1-Year Survival *	2-Year Survival *	3-Year Survival *	4-Year Survival *	5-Year Survival *	Univariate Analysis	Multivariate Analysis	Survival Analysis Outcomes	Clavien–Dindo 0–2 *	Clavien–Dindo 3–5 *
Oshima [20]	NR	NR	NR	NR	NR	HR: 3.2706, *p* = 0.0060	HR: 5.1842 *p* = 0.0036	Higher CAR was associated with significantly shorter overall survival	85% vs. 62%, *p* = 0.3780	15% vs. 28%, *p* = 0.3780
Wijk [21]	84% vs. 87%	67% vs. 73%	50% vs. 60%	34% vs. 46%	NR	HR: 1.406, *p* = 0.028	HR:1.745, *p* = 0.004	Higher CAR is an independent indicator of decreased overall survival	84% vs. 84%, *p* = 1.00	16% vs. 16%, *p* = 1.00
Murakawa [23]	NR	NR	NR	NR	23% vs. 36%, *p* = 0.0089	HR: 1.872, *p* = 0.01	HR: 1.692, *p* = 0.038	Higher CAR was an independent prognostic factor of overall survival	19% vs. 13%, *p* = 0.400	81% vs. 87%, *p* = 0.400
Vujic [23]	56% vs. 67%, *p* = 0.013	29% vs. 35%, *p* = 0.013	12% vs. 21%, *p* = 0.013	8% vs. 15%, *p* = 0.013	3% vs. 7%, *p* = 0.013	OR: 1.454, *p* = 0.036	OR: 1.459, *p* = 0.045	Higher CAR was an independent prognostic factor of overall survival	NR	NR
Ikuta [24]	NR	NR	NR	NR	NR	HR:2.123, *p* = 0.01	HR:1.978, *p* = 0.03	Higher CAR was an independent predictor of poor overall survival	NR	NR
Ikeguchi [25]	NR	NR	NR	NR	NR	NR	OR:2.895, *p* = 0.025	Higher CAR was strong preoperative markers of poor overall survival	NR	NR
Wu [26]	NR	NR	NR	NR	NR	HR: 3.619, *p* = 0.000	HR: 3.99, *p* = 0.000	Higher CAR was an independent and significant predictor of poor overall survival	NR	NR
Haruki [27]	NR	NR	NR	NR	NR	HR: 1.721, *p* = 0.023	HR: 1.726, *p* = 0.035	Higher CAR was an independent and significant predictor of poor overall survival	72% vs. 78%, *p* = 0.519	28% vs. 22%, *p* = 0.519

Abbreviations: HR, Hazard ratio; OR, Odds ratio; CAR; C-reactive protein to albumin ratio; NR, not reported. * High CAR versus Low CAR.

## Data Availability

All evaluated data are available either in the text or the provided tables. The authors did not deal with any additional data.

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
