# Peer review of "Preoperative C-Reactive Protein-to-Albumin Ratio and Its Ability to Predict Outcomes of Pancreatic Cancer Resection: A Systematic Review"

_biomedicines, 2023, doi:10.3390/biomedicines11071983_

Round 1

Reviewer 1 Report

Overall, the manuscript is carefully prepared. The publication summarizes the current role of the C-reactive protein-to-albumin ratio in predicting patients after surgical treatment of pancreatic cancer.

Below are the notes on the systematic review:

1. Authors' affiliations - please systematize the spelling of the country United Kingdom - sometimes there is a full name, sometimes an abbreviation.

2. the introduction is interestingly written, it introduces the reader to the topic well. The purpose of the research was clearly defined.

3. please explain why the study was not registered in PROSPERO.

4. the methodology is correct - I have no criticisms here.

5. authors should present a flow chart of the study.

6. the paper lacks a table summarizing the included studies.

Author Response

Dear Editor,

Many thanks for your kind consideration of our manuscript for revision. Our evidence synthesis group are very grateful to you and the Reviewers for the valid comments which undoubtedly enhanced the quality of our manuscript. Please find below our responses to the Reviewer’s comments.

Reviewer #1:

Overall, the manuscript is carefully prepared. The publication summarizes the current role of the C-reactive protein-to-albumin ratio in predicting patients after surgical treatment of pancreatic cancer.

Our response: We thank Reviewer 1 for the Kind comments.

Below are the notes on the systematic review:

  1. Authors' affiliations - please systematize the spelling of the country United Kingdom - sometimes there is a full name, sometimes an abbreviation.

Our response: We thank Reviewer 1 for the valid comment. We have corrected the Author’s affiliations accordingly.

  1. The introduction is interestingly written, it introduces the reader to the topic well. The purpose of the research was clearly defined

Our response: We thank Reviewer 1 for the Kind positive comments about our introduction

  1. Please explain why the study was not registered in PROSPERO.

Our response: We thank Reviewer 1 for this important query. We now confirm that we have registered the protocol with PROSPERO and provided registration number in the manuscript.

  1. The methodology is correct - I have no criticisms here.

Our response: We thank Reviewer 1 for the Kind positive comments about our methodology

  1. Authors should present a flow chart of the study.

Our response: We thank Reviewer 1 for this important comment. We have provided a Figure 1 as a “Study Flow Diagram”. We have now made it for organised, informative and reader friendly

  1. The paper lacks a table summarizing the included studies.

Our response: We totally agree with Reviewer 1 and in order to address this valid comment we have added an additional Table (Table 4) to summarise the included studies.

Looking forward to your favourable consideration

Yours sincerely

Reviewer 2 Report

Dear colleagues

The following article entitled “Preoperative C-reactive protein-to-albumin ratio and its ability to predict outcomes of pancreatic cancer resection: a systematic review” is well written. The manuscript stated the preoperative CAR value seems to be an important prognostic score in predicting survival outcomes in patients undergoing pancreatic cancer resection. However, the data show a lack of knowledge of the sensitivity and specificity of CAR. Many papers have been previously investigated the relationship between CRP and prognosis in various types of cancer including pancreatic cancers. Thus, please, clarify the main points in the present study ensuring and highlighting the novelty. Please add relation of CAR with other cytokines should be addressed in the discussion.

Author Response

Dear Editor,

Many thanks for your kind consideration of our manuscript for revision. Our evidence synthesis group are very grateful to you and the Reviewers for the valid comments which undoubtedly enhanced the quality of our manuscript. Please find below our responses to the Reviewer’s comments.

Reviewer #2:

Dear colleagues

The following article entitled “Preoperative C-reactive protein-to-albumin ratio and its ability to predict outcomes of pancreatic cancer resection: a systematic review” is well written. The manuscript stated the preoperative CAR value seems to be an important prognostic score in predicting survival outcomes in patients undergoing pancreatic cancer resection. However, the data show a lack of knowledge of the sensitivity and specificity of CAR.

 Our response: We thank Reviewer 2 for the Kind comment about our manuscript and careful evaluation of the findings

  1. Many papers have been previously investigated the relationship between CRP and prognosis in various types of cancer including pancreatic cancers. Thus, please, clarify the main points in the present study ensuring and highlighting the novelty.

Our response: We are grateful to Reviewer 2 for this valid recommendation.  Following this valid comment, we have added the following to

Introductionà paragraph 4

“A novel and robust systematic review should focus on only preoperative value of CAR in resectable pancreatic cancer, assess the reliability of the provided cut-off values reported by the best available evidence, and provide strategies to improve the quality of the evidence in order to prepare CAR for potential inclusion in future prediction models for pancreatic cancer. We aimed to conduct a comprehensive systematic review of all available comparative studies to evaluate the ability of CAR in predicting outcomes in patients undergoing pancreatic cancer resection”

Discussionà paragraph 6

This study is the first review that comprehensively evaluated the best available evidence surrounding ability of CAR to predict the outcomes of pancreatic cancer resection and recognized the shortcomings of such evidence. Considering the findings of this review, we have the following novel directions for future research to enhance the quality of evidence and provide appropriate grounds for higher level evidence…

  1. Please add relation of CAR with other cytokines should be addressed in the discussion.

Our response: We totally agree with Reviewer 2’s comment. Unfortunately, the available literature around the other cytokines and inflammatory markers are very limited particularly when no definitive relation between those or CAR have been investigated to the best of our knowledge. Nevertheless, we believe CAR and Neutrophil to lymphocyte  ratio (NLR) have the best potential to be concentrated on. Therefore, in order to address this valid comment, we have added the followings to:

Discussionàparagraph 6

“Several other cytokines or inflammatory markers can potentially be considered as representatives of cancer induced inflammation in patients with pancreatic cancer. Nevertheless, the available literature around other inflammatory markers are even more limited than CAR. One of the advantages of CAR is that its components are available in routinely reported blood results of almost all patients presenting to secondary/tertiary care. Among inflammation-based prognostic scores, CAR and NLR have the greatest potential to represent the degree of inflammation in any future predictive model.”

Looking forward to your favourable consideration

Yours sincerely

Round 2

Reviewer 1 Report

The authors have addressed my remarks in a thorough and satisfactory fashion.

Reviewer 2 Report

I am totally accepting the manuscript in the present form.